# Shear Wave Velocity Determination of a Complex Field Site Using Improved Nondestructive SASW Testing

**DOI:** 10.3390/s24103231

**Published:** 2024-05-19

**Authors:** Gunwoong Kim, Sungmoon Hwang

**Affiliations:** 1Department of Geotechnical Engineering Research, Korea Institute of Civil Engineering and Building Technology, Goyang 10223, Republic of Korea; 2Department of Civil, Architectural, and Environmental Engineering, University of Texas at Austin, Austin, TX 78712, USA; sungmoon@utexas.edu

**Keywords:** nondestructive testing, SASW, seismic testing, surface wave, Vs profile

## Abstract

The nondestructive spectral analysis of surface waves (SASW) technique determines the shear wave velocities along the wide wavelength range using Rayleigh-type surface waves that propagate along pairs of receivers on the surface. The typical configuration of source-receivers consists of a vertical source and three vertical receivers arranged in a linear array. While this approach allows for effective site characterization, laterally variable sites are often challenging to characterize. In addition, in a traditional SASW test configuration system, where sources are placed in one direction, the data are collected more on one side, which can cause an imbalance in the interpretation of the data. Data interpretation issues can be resolved by moving the source to opposite ends of the original array and relocating receivers to perform a second complete set of tests. Consequently, two different Vs profiles can be provided with only a small amount of additional time at sites where lateral variability exists. Furthermore, the testing procedure can be modified to enhance the site characterization during data collection. The advantages of performing SASW testing in both directions are discussed using a real case study.

## 1. Introduction

The spectral analysis of surface waves (SASW) method is a nondestructive seismic technique that takes advantage of the dispersive features of Rayleigh-type surface waves traveling along the surface of a layered half-space to determine the shear wave velocity profile. The SASW method was originally introduced in the 1980s [1,2]. The Rayleigh-type surface waves travel past pairs of vertical receivers that measure phase velocities throughout a broad spectrum of frequencies, resulting in a large variety of wavelengths. The data reduction procedure in this study was conducted by utilizing WinSASW software (ver 4.1.5), developed by Joh [3]. The collected data at the field site were filtered through WinSASW software to generate a dispersion curve and then determine the Vs profile that fit the dispersion curve, which represented the site features. SASW does not require intrusive boring, which makes it more cost-effective than other geotechnical site investigation tests, and provides Vs values that can be used in seismic designs. In addition, SASW has been utilized for decades at a variety of sites, including normal ground sites, as well as tunnels, dams, and other structures due to its ability to be tested in a variety of site conditions [4,5,6,7] and for many different purposes [8,9,10].

A reliable geotechnical investigation is crucial to reduce the time and financial risks of a construction project [11]. At complex sites, proper geotechnical information often becomes difficult to obtain, and many studies have been conducted to overcome these challenges. For instance, point-based methods, such as SPT and CPT, only cover a very small area; therefore, in order to determine the actual geotechnical investigation, many tests may involve significant expenses. In practical terms, performing many tests brings significant financial and time challenges. Thus, several studies have been conducted to minimize these concerns. For example, studies have been conducted using kriging or interpolation methods to predict missing geotechnical information [12,13,14,15] or find optimal test locations to reduce the number of tests [16]. While SASW tests appear to provide accurate resolution at shallow depths [17], there is an issue with resolution at deeper depths since, unlike point-based tests, SASW tests provide averaged measurements over a broad area. Statistical techniques have been proposed to enhance the resolution in SASW analyses [18]. Scholars have long emphasized the importance of accurate and precise geotechnical investigations. Therefore, this paper also presents a partial solution to improve the resolution of the SASW test, a nondestructive test.

Traditionally, SASW collects data at various depths by placing an energy source at one side of the testing array and gradually increasing the distance between the receiver and the source. This method provides a convenient and inexpensive way to acquire reliable data. However, the traditional procedure may have a limitation when evaluating sites with lateral variability, and due to the original configuration of the test, only one direction can be deeply profiled [19]. 

Thus, this study discusses an improved SASW field testing procedure, whereby a second complete set of tests is conducted in the reverse direction to overcome the limitations of traditional SASW testing. The improved SASW test provides two solutions for a laterally complex site instead of one, allowing for a higher-resolution analysis considering lateral variability. The improved method overcomes a limitation of the traditional method by enabling depth-profiling in both directions. This novel technique provides the opportunity for determining the same depth-profiling in two opposing directions, thereby promoting the accuracy and precision of the results. Notably, the extra time required to acquire twice the data takes only 25–30% of the time compared to the traditional method. These additional data enhance the reliability of the test results and facilitate reviewing of the test during the process. 

The rest of the paper is structured as follows. First, the conventional SASW testing method is introduced. Then, the improved SASW testing method is presented. The advantages of the improved method are then summarized. Lastly, the improved SASW test is validated using a case study.

## 2. Traditional SASW Testing

### 2.1. Traditional SASW Field Testing Procedure

The multi-layered, half-space represented in Figure 1 is utilized to illustrate the dispersion characteristics of Rayleigh wave velocity. The SASW method employs the Rayleigh waves that propagate along the surface. The propagation of high-frequency, shorter wavelength surface waves and low-frequency, longer wavelength surface waves is illustrated in Figure 1. The shorter wavelength samples data near the shallow surface and the longer wavelength samples data from deeper depths to shallow depths. Combining all the wave information from the various wavelength leads to the dispersion curve that represents the testing site. 

SASW testing utilizes Rayleigh-type waves generated by using a hammer or vibroseis, depending on the specific requirements of the site investigation [20]. The vertical movement of this wave along the line of the array is captured simultaneously utilizing pairs of receivers. The standard testing arrangement consists of three receivers and one source, as shown in Figure 2. In this arrangement, the middle receiver (R2) is located at the centerline of the test array and remains fixed during the entire testing procedure. The distance (x) between receivers R1 and R2 is maintained at the same length as the distance (x) between the receiver and the source. Simultaneously, the middle receiver (R2) makes a pair with the other receiver (R3). The distance between the source and middle receiver is 2x, which is the same distance between the far receiver (R3) and the middle receiver (R2). This setup allows for both measurements of data from two different spacings (R1–R2 and R2–R3). The test is performed by increasing the distance between the source and the three receiver pairs while maintaining the distance ratio. In general, an SASW test can collect data at a depth proportional to the distance of the receiver pair. Unless limited by source energy, testing is performed at the intended profiling depth of the site by increasing the array length.

### 2.2. Development of the Vs Profile

After data collection at the site, the first phase of the SASW analysis process involves generating a field dispersion curve. An example is provided to illustrate this procedure, using data obtained at the site with a receiver spacing of 45.7 m (150 ft) (Figure 3). As shown in Figure 3a, data below 5 Hz were not acquired because this portion of the data is part of the near-field zone (0 to 180 phase degree) and is not utilized for data analysis. After masking the near-field portion of the collected data, the data can be unwrapped using WinSASW software to generate the dispersion curve by utilizing the following relationship [21,22]:(1)$$V_{R}=f\;\times \;\lambda =f\;\times \;(360{}^{\circ}/Ø)\;\times \;d$$
where,

f = Frequency;λ = Wavelength;d = Receiver Spacing;Ø = Unwrapped Phase Angle. 

Based on Equation (1), the lowest-frequency data correspond to the longest wavelength data (#1 in Figure 3b), and the highest-frequency data correspond to the shortest wavelength data (#4 in Figure 3b). As can be seen in this figure, the lower-frequency data represent longer wavelengths, i.e., phase velocity information at greater depths. Unwrapping all data from various spacings collected in the field through this process produces the experimental (field) dispersion curve (Figure 4) that contains the characteristics of the site [23].

**Figure 3 sensors-24-03231-f003:**
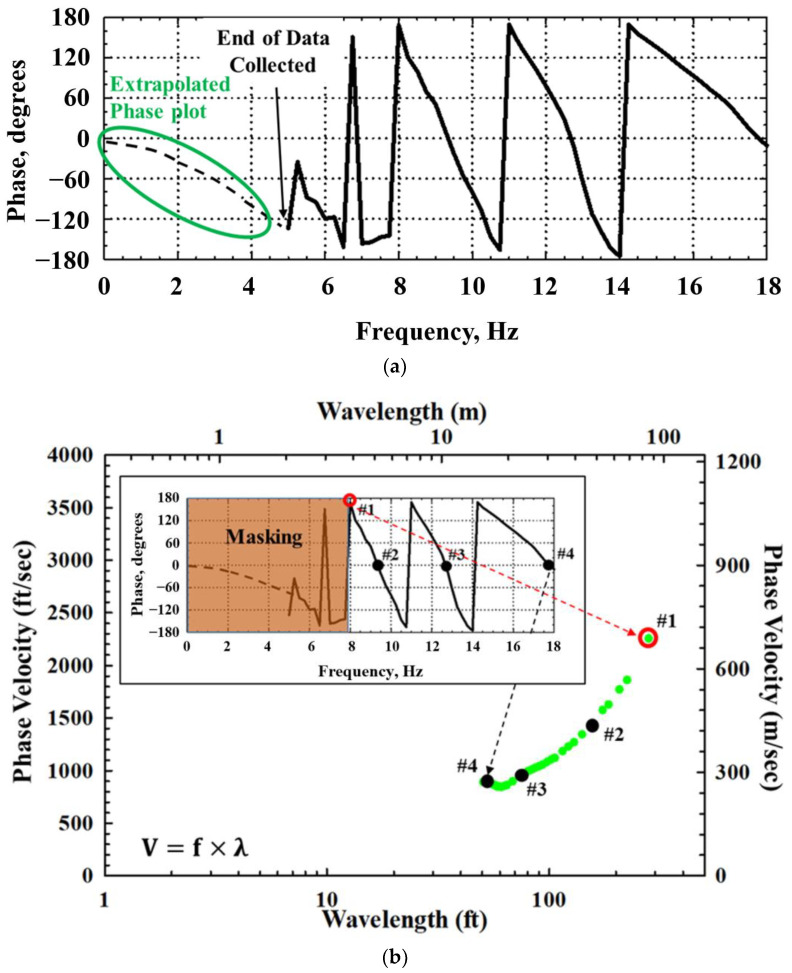
Development of the field dispersion curve generated from one receiver spacing (45.7 m (150 ft)): (**a**) wrapped phase plot measured from 45.7 m (150 ft) receiver spacing; and (**b**) individual experimental dispersion curve generated from masked phase plot (45.7 m (150 ft)) receiver spacing.

The experimental (field) dispersion curve in Figure 4 was generated utilizing data from 10 different receiver spacings. As shown in the figure, the longest receiver spacing was 45.7 m, which resulted in a dispersion curve with wavelengths of about 90 m. This dispersion curve highlights two advantages of the SASW testing, as follows: (1) The overlap of dispersion data acquired from the various spacings allows for cross-checking during data analysis. Cross-checking allows for real-time validation during the analysis process, which adds credibility to the data analysis; Another benefit is that (2) the analysis can be carried out for sites with inverse stiffness of the soil. In Figure 4, the highest velocities appeared in the lower-wavelength range, i.e., at shallow depths, reflecting the presence of rigid asphalt and concrete layers on the site surface. In this case, SASW has a capability to characterize a soil layer where there is an inversion of stiffness with increasing depths.

The last step was to develop a Vs profile from the created experimental (field) dispersion curve (Figure 5). In this process, a compacted dispersion curve, representing the experimental (field) dispersion curve, is generated by computing a 4th order moving average (Figure 5a). Then, the theoretical dispersion curve that best fit the compacted dispersion curve was matched (Figure 5b) through a forward modeling process or inversion analysis [24,25]. In this step, the stiffness matrix [26] is applied to derive the Vs profile. Finally, the Vs profile were determined that represented the geological information of the testing array (Figure 6). As can be seen from the process described above, SASW requires several steps to perform an analysis. Recently, studies have been conducted on automated analysis using machine learning or secondary indicators to make the analysis more convenient [27,28].

## 3. Improved SASW Testing

### 3.1. Benefit of Improved SASW Field Testing

The traditional SASW approach offers a convenient and cost-effective means of obtaining reliable data. However, the original technique may be limited when determining sites with lateral variability. Furthermore, only half the side of the testing array can be deeply profiled in the original set-up. The improved SASW testing method can simply overcome these limitations by performing extra testing. The improved SASW measurements involve performing the test in both the forward and reverse directions. Performing the test in both directions increases the test duration by about 25~30%, but the benefits outweigh the additional time spent. There are two main benefits of performing the tests in both directions. First, the lateral variables at the site can be identified during and after the data collection at the site. Figure 7a shows an example of collecting data with the same distance to the left and right of center in a test array. The red block shows the data on the right side of the center, and the yellow block shows the data on the left side of the center (Figure 7b). Both data exhibit the same pattern in the high-frequency range, but the data pattern starts to differ in the low-frequency range. The difference between the left and right sides in the low-frequency range from the center indicates that the deeper the layer, the greater the difference in ground stiffness between the right and left sides. Additionally, the testing procedure can be modified to improve the site characterization if the differences in two directions are detected during data collection. Consequently, it is possible to determine the differences at sites where lateral variability exists.

The second advantage of the improved SASW method is that deep profiling can be performed on both sides of the centerline. Figure 8 shows the area of data collection using a test array with a longest spacing of 30 m. The gray areas show the area of data collection in the first (2, 4 m spacing) and second (5, 10 m spacing) sets of the test, and the blue and green areas show the area of data collection in the third set (15, 30 m spacing) of the test. As shown in the figure, when the test was performed in the traditional way (Figure 8a), only one side of the centerline (0 to 30 m) could be deeply profiled. However, when the test is additionally performed in the opposite direction (0 to −30 m) as shown in Figure 8b, both sides can be deeply profiled (Figure 8c). Furthermore, as the overlapping areas increases, the reliability of the data analysis increases.

The improved SASW test provides two solutions based on the center, whereas the traditional SASW test provides one solution from a single array. However, the improved test divides the left and right sides based on the position of the center receiver, which makes it dependent on the location of the center receiver where the test is performed. One possible solution to address this is adding a receiver between the center and far receivers. This approach divides the array into four sections, leading to higher resolution.

### 3.2. Validation of Improved SASW Field Testing through Case Study

In this section, a case study of SASW testing on the crest of a dam is presented. The nondestructive SASW test was performed to assess the condition of the embankment dam located in the northwestern part of the United States. The use of nondestructive testing is critical, as traditional investigation methods were often intrusive and interrupted the functioning of structures [29]. The dam comprises roughly 15 m of embankment fill and alluvium material, while the underlying bedrock comprises either basalt or vesicular basalt. The SASW test was conducted at the far end of the dam, which is a complicated location influenced by both the bedrock below the dam and the sloping bedrock on the landward side. Due to the complexity of the geological formation, an improved SASW test, with a resolution higher than the traditional test, was performed.

This case validates the improved SASW method described in the previous section. Due to the complex ground geology at the site, two different dispersion curves were generated from a single SASW array (Figure 9). The experimental dispersion curves indicate a phase velocity difference of about 11% for wavelengths longer than 30 m (100 ft). On the right side of the array, the phase velocity at a wavelength of 100 feet is approximately 335 m/s (1100 ft/s), while on the left side, it is about 375 m/s (1230 ft/s). Consequently, two distinct dispersion curves were separated to assess the lateral variability on this site. The difference in wave velocities continued to increase as the wavelength increased. In this example set of an SASW array, the maximum receiver spacing used in the field is 45.7 m (150 ft). The maximum spacing is important because the longest wavelength dispersion data that can be used without including near-field data are 2 × 45.7 = 91.4 m (300 ft). Therefore, the maximum depth of the Vs profile is 45.7 m (150 ft), which is half of the maximum wavelength. Due to the lateral variability at this location, two distinct dispersion curves were determined. This variability could be detected because SASW testing involved testing in both the forward and reverse directions along the linear array.

In Figure 10a, the experimental dispersion data that were collected for testing performed from the centerline to the left side (black dot in Figure 9) of the array are presented and the experimental dispersion data that were collected for testing performed from the centerline to the right side (green dot in Figure 9) of the array are shown in Figure 10d. The two separate dispersion curves follow the process described in Section 2. The Vs profile is created by: (1) generating a compacted dispersion curve from the experimental dispersion curve (Figure 10b,e); and (2) finding the theoretical dispersion curve that best matches the compacted dispersion curve (Figure 10c,f).

Finally, two Vs Profiles were determined from two distinct dispersion curves (Figure 11). Similar to the dispersion curve, the center to the left side has relatively stiffer bedrock at shallower depths than the center to the right side. The difference in the Vs values on both sides of the centerline increases with the depth, and the difference in Vs values became more than 10% at about 15 m (50 ft). A 10% difference between the two Vs values equals about a 20% difference in shear modulus. Therefore, the dispersion curve should be divided into two and analyzed separately. 

To validate the Vs profile obtained from the SASW test, geologic information was used to compare the values. The two Vs profiles determined using the SASW testing showed good agreement with the geologic information that was provided (Figure 12). The area utilized to acquire data through the SASW test has a triangular shape, as shown in Figure 8. Therefore, for a proper comparison, the triangular shape and Vs profile were compared to scale. Based on the site information, there is shallower bedrock located in the on the left side of the array due to the sloping shallow bedrock layer from the mountain. Obviously, the bedrock depth of the left side of the centerline (13 m (43 ft)) is shallower than the right side (16 m (52 ft)). Moreover, the Vs value of basalt (1097 m/s (3600 ft/s)) shows a higher value than the mixture of vesicular basalt and clay layer (853 m/s (2800 ft/s)), which seems reasonable.

The Vs values of the corresponding materials at similar depths obtained from other sites were compared to validate the Vs values obtained from the case study (Table 1). The data used for comparison were acquired at similar depths to avoid the influence of depth-dependent confining effects. The Vs value for the embankment fill layer showed 229–381 m/s (750–1250 ft/s) at this site with no lateral difference. These values were similar to the Vs values of 244–396 m/s (800–1300 ft/s) for embankment fill layers at another dam. The alluvium material was not detected on the left side due to the thin layer. Although the thickness slightly differed from the thickness shown in the existing geologic information, an alluvium material was detected on the right side. The Vs value of the alluvium material at this site was 335 m/s (1100 ft/s), within the range of values found at another dam site. Lastly, the bedrock showed a Vs value of 1097 m/s (3600 ft/s) for the left side consisting of basalt only, which lies in the range of other sites, and 853 m/s (2800 ft/s) for the right side consisting of basalt and clay, which again lies in the range of other sites.

In summary, the case study demonstrated the reliability and robustness of the improved SASW method in distinguishing the Vs values of the left and right sides with different depth ranges. Without this method, only one side would have been depth-profiled, potentially leading to a biased representation of the site depending on the direction of the test. For instance, if the traditional method was used with the source on the right side, it would have been challenging to identify the presence of the alluvium material or the relatively weak vesicular basalt with a clay layer.

## 4. Discussion and Conclusions

The improved SASW method was studied, particularly with respect to the data collection process and the data interpretation strategy based on the improvement. The SASW is a noninvasive seismic method that uses the dispersive characteristics of Rayleigh-type surface waves to determine the shear wave velocity profiles at geotechnical sites. The most important aspect of the improved SASW method is the performance of a second set of tests in the opposite direction to the traditional SASW method. Traditionally, the SASW testing is performed only in one direction (forward direction) to determine the Vs profiles at the site. However, additional testing in the opposite (reverse) direction provides complete profiling on both sides of the array. If it exists, the lateral variability in Vs is detected by testing in both the forward and reverse directions along the entire length of the array. The key findings of this study are summarized as follows:(1)Additional testing in the reverse direction enabled not only deeper profiles on both sides of the array but also increased data reliabilities and resolution due to extra overlapping data. In practice, performing additional tests in the opposite direction requires about 25–30% extra time and effort but provides the advantage of obtaining twice as much data;(2)Another benefit of additional testing in the opposite direction includes the availability of checking the SASW data during collection. If a difference in data with the same spacing is found during a test, the test procedure can be adjusted to improve the quality of the site characterization. In practice, the quality improvement of data collected during the field test results in a significant improvement in the accuracy of data analysis;(3)The testing and data-analysis procedures of the improved SASW method were explained by showing a dataset from a real, laterally variable field site. Two main differences were found in this real dataset. First, the left side of the centerline showed higher Vs values in the bedrock. Second, the depth of the bedrock on the left side of the centerline is shallower than the right side. The original geologic information was used to validate these two differences found from the SASW test. Based on the investigation, these two lateral differences exist at the site. It would not be possible to see the differences between the two sides from the centerline if traditional SASW was performed.

The improvements in the SASW method allow for better analysis of complex geotechnical sites and enhanced accuracy. Although this study improves on the traditional method, the improved method is still affected by the center receiver location as the length of the array increases. As a solution, the test can even be improved by installing an additional receiver between the center and the far receiver, dividing sections into four instead of two. Future experiments with additional geophones between the arrays are expected to provide more precise analyses.

## Figures and Tables

**Figure 1 sensors-24-03231-f001:**
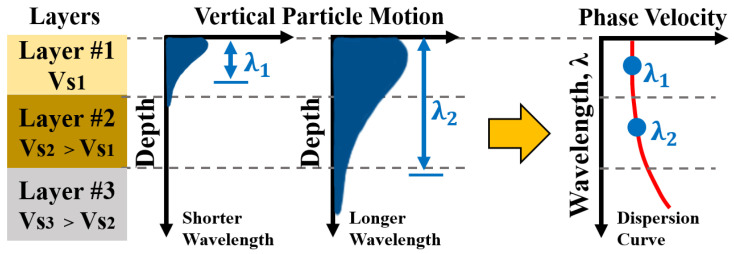
Surface waves with different wavelengths (λ1 and λ2) sampling a layered system.

**Figure 2 sensors-24-03231-f002:**
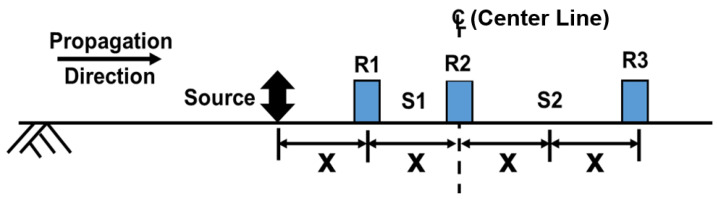
Generalized SASW field arrangement.

**Figure 4 sensors-24-03231-f004:**
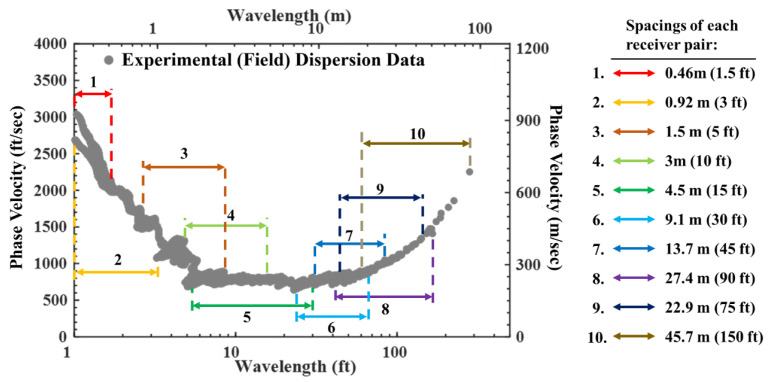
Experimental (field) dispersion curve generated from all 10 receiver spacings.

**Figure 5 sensors-24-03231-f005:**
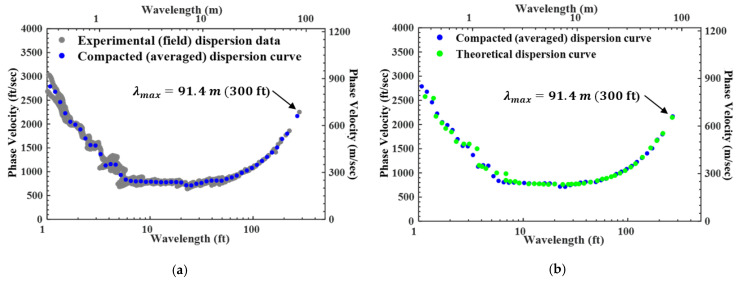
SASW data analysis procedure: (**a**) experimental (field) and compacted dispersion curve, and (**b**) comparison of the compacted and theoretical dispersion curve.

**Figure 6 sensors-24-03231-f006:**
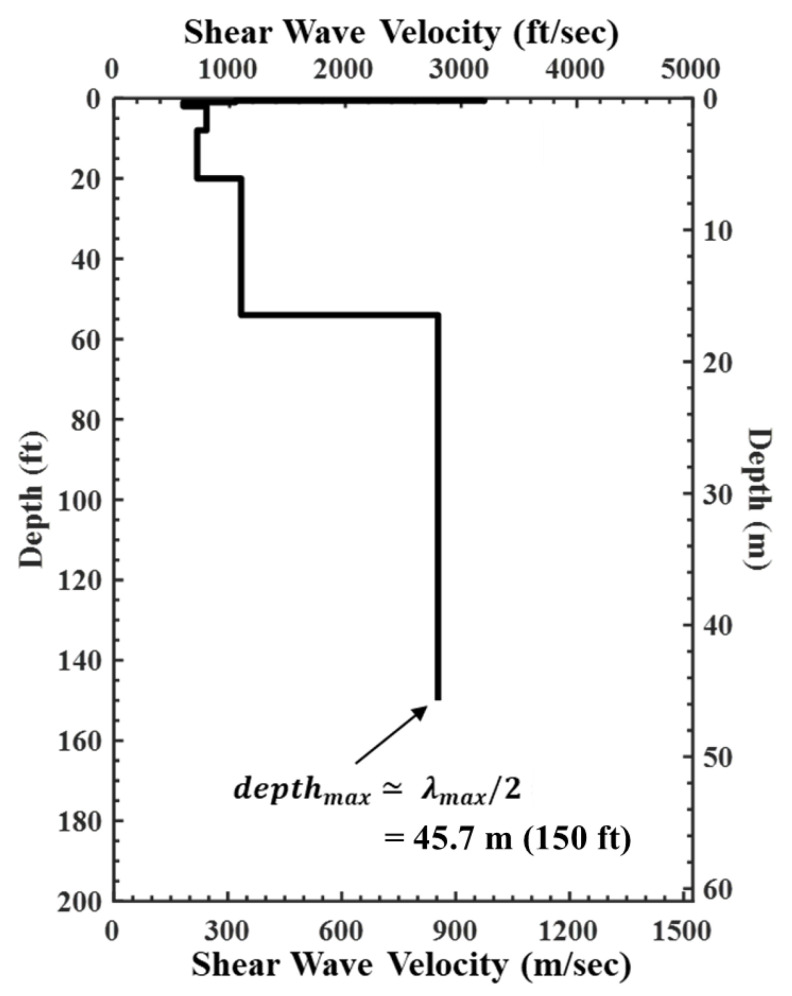
Resulting Vs profiles from fitting the compacted dispersion curve in Figure 5b.

**Figure 7 sensors-24-03231-f007:**
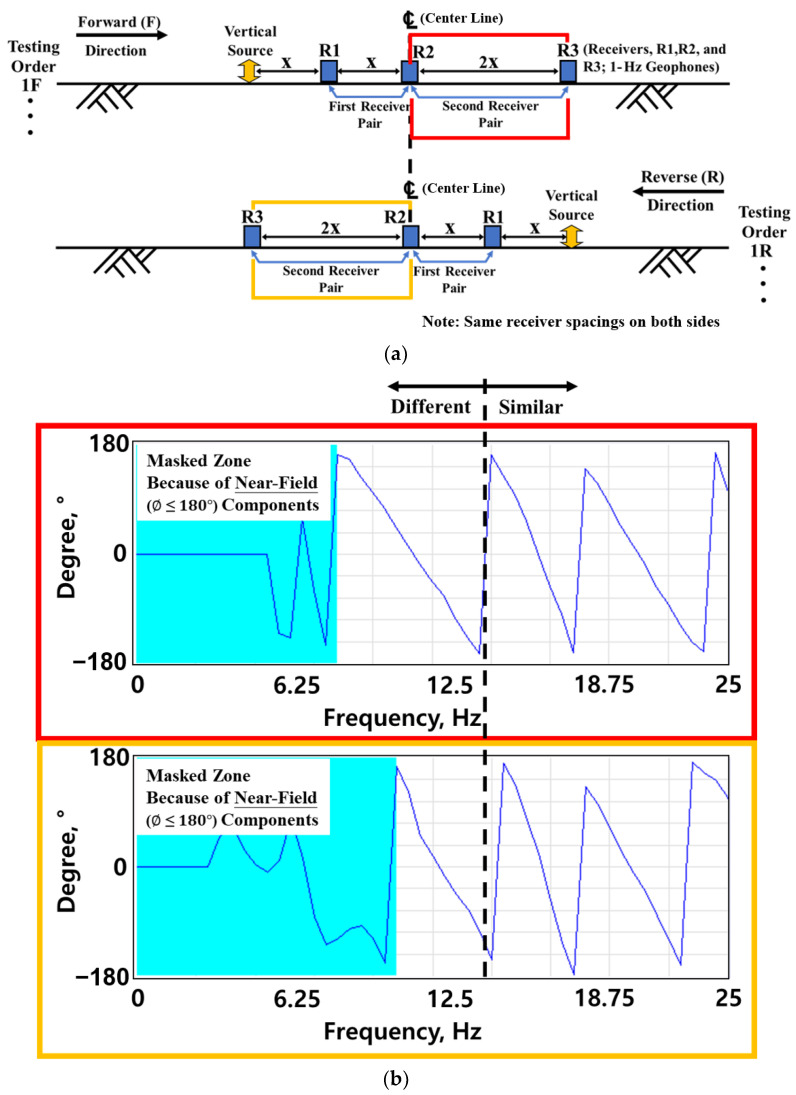
Detection of the lateral variabilities during the SASW data collection: (**a**) SASW field arrangement in forward and reverse directions; and (**b**) example comparison of masked wrapped phase plot measurement from the same receiver spacings in forward and reverse directions at the laterally variable site.

**Figure 8 sensors-24-03231-f008:**
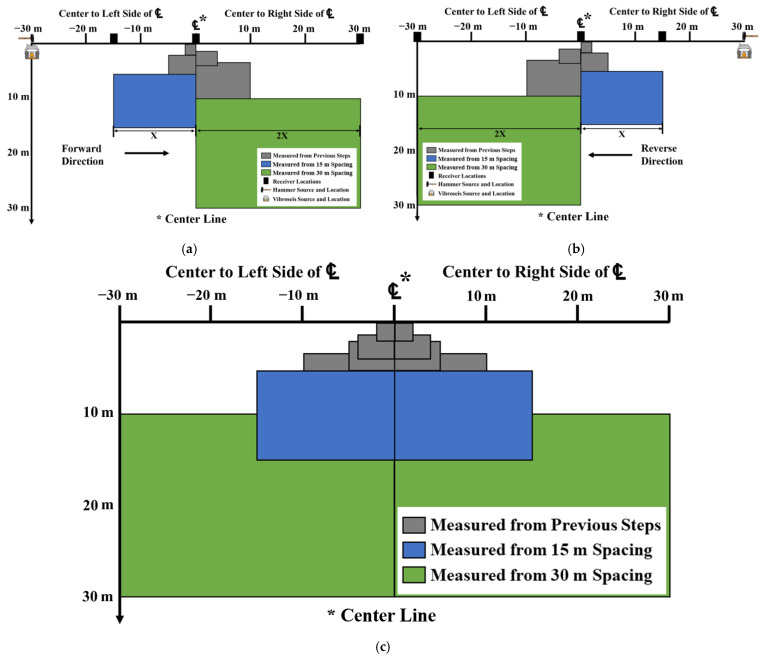
Comparison of the areas of data collected with the traditional and improved SASW methods: (**a**) forward direction (original); (**b**) reverse direction; and (**c**) both directions (improved).

**Figure 9 sensors-24-03231-f009:**
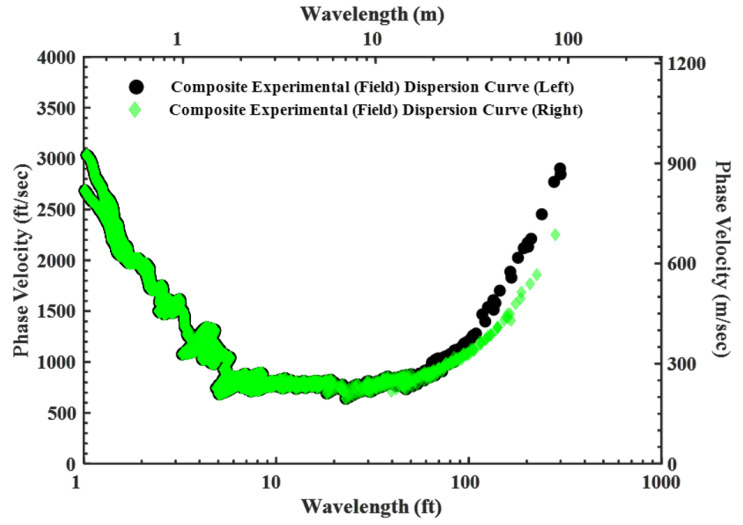
Determination of lateral variability using improved SASW testing method.

**Figure 10 sensors-24-03231-f010:**
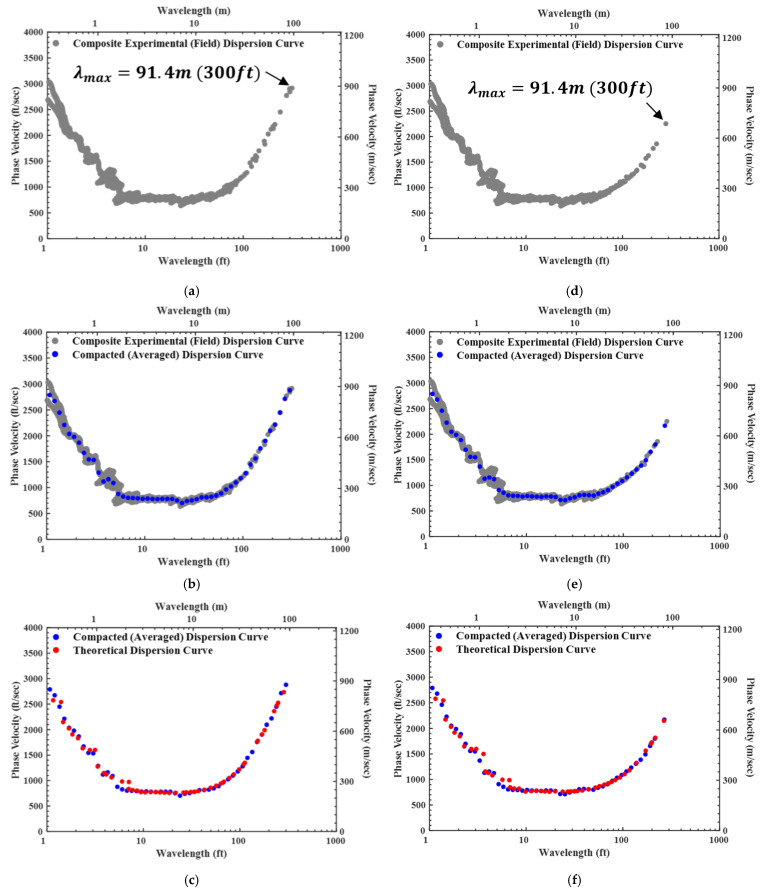
SASW data analysis processes using the dispersion data from Figure 8: (**a**) experimental dispersion curve (**left** side); (**b**) experimental and compacted dispersion curves (**left** side); (**c**) compacted and theoretical dispersion curves (**left** side); (**d**) experimental dispersion curve (**right** side); (**e**) experimental and compacted dispersion curves (**right** side); and (**f**) compacted and theoretical dispersion curves (**right** side).

**Figure 11 sensors-24-03231-f011:**
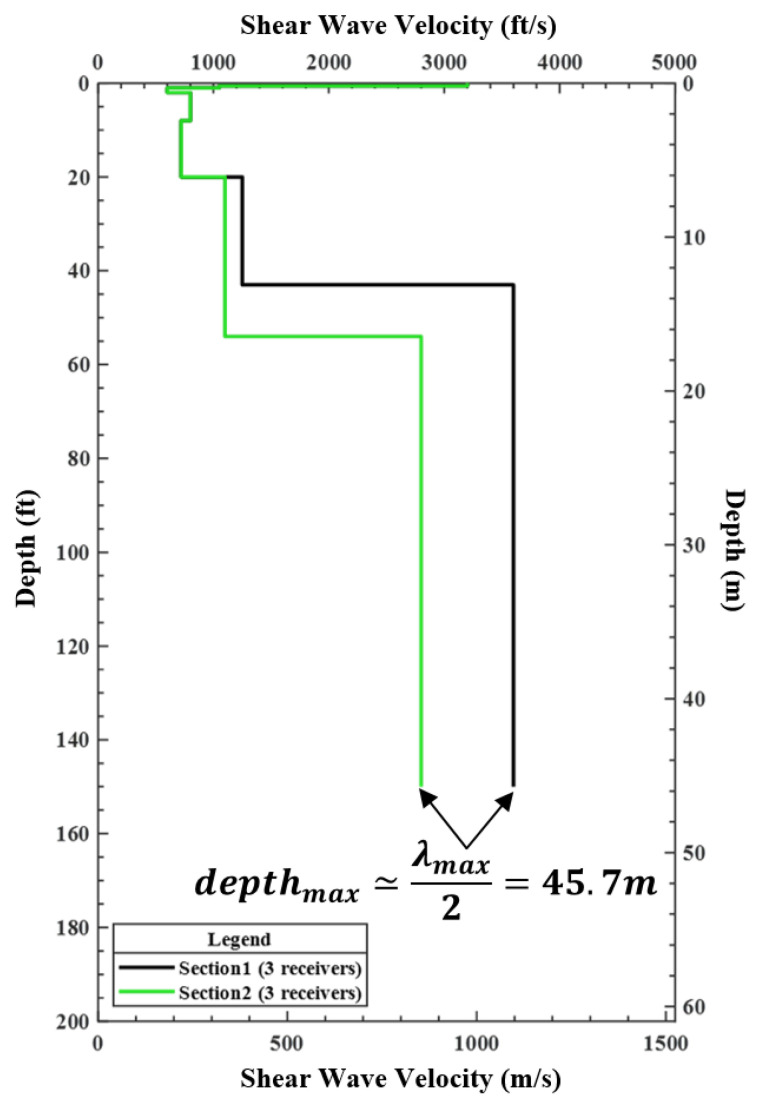
Comparisons of two Vs profiles determined at the site.

**Figure 12 sensors-24-03231-f012:**
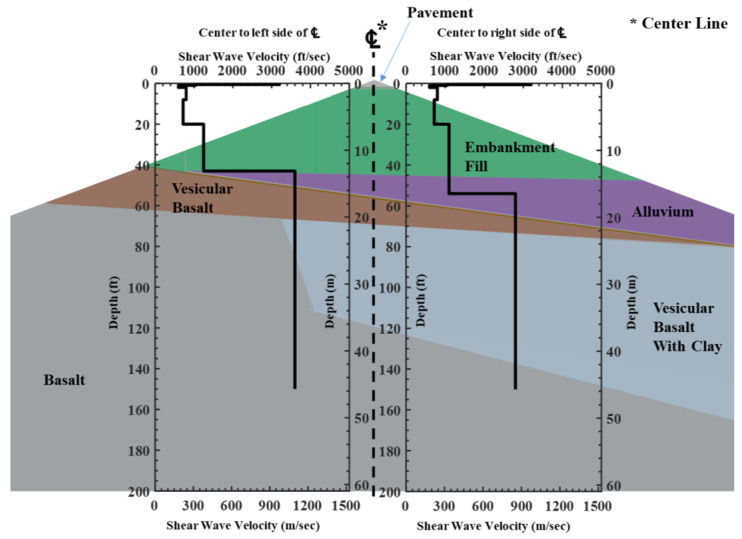
Comparison of the original geologic information and SASW Vs profiles.

**Table 1 sensors-24-03231-t001:** Comparison of the shear wave velocity and depth ranges that were sampled from other sites.

Material Type (Depth Range)	This Site (Left Side)	This Site (Right Side)	Site 1 (Dam)	Site 2 (Field Site)	Site 3 (Field Site)
Embankment Fill (1~40 ft) (0.3~12 m)	750~1250 ft/s (229~381 m/s)	750~1100 ft/s (229~335 m/s)	800~1300 ft/s (244~396 m/s)	-	-
Alluvium (40~60 ft) (12~18 m)	-	1100 ft/s (335 ms)	1000~1600 ft/s (305~488 m/s)	-	-
Vesicular, Aphanitic, and Hard Basalt (50~150 ft) (15~46 m)	3600 ft/s (1097 m/s)	-	2900~3500 ft/s (884~1067 m/s)	-	3000~4000 ft/s (914~1219 m/s)
Basalt with sediment (Silt, Clay) (50~150 ft) (15~46 m)	-	2800 ft/s (853 m/s)	-	2600~2800 ft/s (792~853 m/s)	-

## Data Availability

The original contributions presented in this study are included in the article, and further inquiries can be directed to the corresponding author.

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
