# Peer review of "Shear Wave Velocity Determination of a Complex Field Site Using Improved Nondestructive SASW Testing"

_sensors, 2024, doi:10.3390/s24103231_

Round 1
Reviewer 1 Report
Comments and Suggestions for Authors
(1) I think the title can be modified to better reflect the main contribution of the study. There is no Stiffness Determination contents in the paper. It is better to remove these two words in the title.
(2) The abstract can be greatly improved. In its current form, there is too much details on the basics of traditional SASw technique. It is better to focus on the solutions to overcome traditional limitations, the proposed methodology, the benefits of the present work.
(3) The novelty and scientific contribution of the present work should be clearly indicated in the Introduction section. Please clearly describe the proposed methodolog, the main contribution over traditional apporach, and the expected benefits.
(4) In section 3.2, please give more details on the case study of SASW testing on the crest of a dam. Where is the dam located? What are the geology imformation?
(5) Discuss the limitations of proposed approach?
Author Response
Thank you very much for taking the time to review this manuscript.
Please see the attachment.

Reviewer 2 Report
Comments and Suggestions for Authors
The manuscript is well presented. It is not something ground shaking, but it can make a useful publication.
I have some comments:
The presentation of graphs and figures must be improved. They are very small, some of them of low quality/resolution etc.
In order to understand the reader, you should add more quantitative information on the presented case study and discussion regarding the benefits in terms of accuracy and effectiveness.
The introduction should include more systematic information regarding the scope, objectives and innovation of the presented research.
I believe 10.3390/s21010314 can be useful for the introduction improvement.
Comments on the Quality of English LanguageMinor
Author Response

(The authors gave the same response as above.)

Round 2
Reviewer 1 Report
Comments and Suggestions for Authors
The authors have addressed my comments.
Reviewer 2 Report
Comments and Suggestions for Authors
Accept
Comments on the Quality of English LanguageMinnor